# Development of Virtual Flow-Meter Concept Techniques for Ground Infrastructure Management

**Ruslan Vylegzhanin [1], Alexander Cheremisin [2,*], Boris Kolchanov [1], Pavel Lykhin [1], Rustam Kurmangaliev [1], Mikhail Kozlov [1,3], Eduard Usov [3,4]** and **Vladimir Ulyanov [1,3]**

1. Novosibirsk R&D Center LLC, 630090 Novosibirsk, Russia
2. Skolkovo Institute of Science and Technologies, 121205 Moscow, Russia
3. Physical Department, Novosibirsk State University, 630090 Novosibirsk, Russia
4. Novosibirsk Branch of the Nuclear Safety Institute of the Russian Academy of Sciences, 630090 Novosibirsk, Russia
* Correspondence: a.chermeisin2@skoltech.ru

**Abstract:** This paper describes the further development of the virtual flow meter concept based on the author's simulator of an unsteady gas–liquid flow in wells. The results of comparison with commercial simulators based on real well data are given as practical applications. The results of the comparison of the simulators demonstrated high correspondence (<10% error) for a number of target parameters. The description of the architecture and results of testing the algorithm for automatic settings of the model parameters are given. Operating speed was the key criterion in the architecture development. According to the test results, it became possible to achieve the adaptation accuracy of 5% specified.

**Keywords:** virtual flow measurement; multiphase flow; reservoir fluid; simulator; numerical methods

## 1. Introduction

The so-called digital-twin technologies designed to offer a commercial solution to a number of process problems are being actively implemented in many industrial areas.

A simulator of an unsteady flow of a multiphase fluid in a reservoir, a well, and surface gathering lines, which takes into account all the key physical processes occurring during hydrocarbon production, is actually used as a twin for oil and gas fields. This paper contains a brief description of the key features of the unsteady multiphase flow simulator developed by the authors taking into account the choke and electric centrifugal pump models.

Correlating the data obtained during the simulation with the actually measured data is a necessary step when using a simulator in a digital twin. This process is called adaptation or adjustment to an actual value.

The simulator parameters can be grouped according to two criteria: impact on the result, i.e., sensitivity, and accuracy of their measurement in a real field.

The figure (Figure 1) shows some parameters for the black oil model. Adaptation consists of selecting highly sensitive and weakly measured parameters so that the input parameters sent to the simulator provide the output ones corresponding to the measured ones with sufficient accuracy for a wide set of input data. Gas viscosity, gas compressibility, and pipe roughness were used as adjustment parameters.

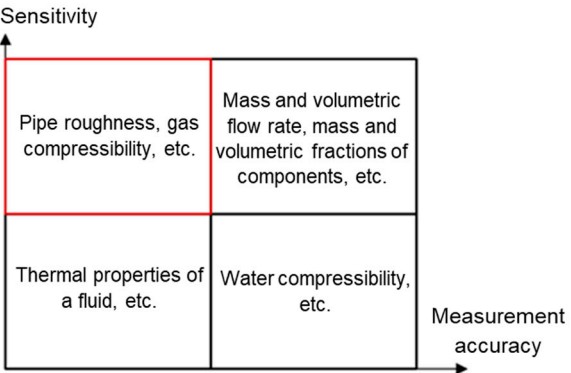

**Figure 1.** Simulator parameters grouped by the two criteria.

It should be noted that it is incorrect to equate this process with the calibration of an instrument because the development of an instrument concept consists in minimizing the number of highly sensitive and weakly measured parameters, while calibration consists of selecting highly sensitive and accurately measured parameters.

Two neural networks were used as the adaptation algorithm of the simulator. The first was used to make a forecast of the simulator output parameters based on the input parameters.

The second was used to assess the optimal input parameters for the given output parameters.

The neural network implementations were taken from the TensorFlow software library.

## 2. Methodological Approaches

### 2.1. Mathematical Model of an Unsteady Flow of a Multiphase Fluid

The proposed simulator uses the systems of equations of continuity, conservation of mass, energy, and momentum (for the gas and liquid phases) [1–4], as well as the equations for the total enthalpy of a mixture in the allocated volume and for the mass of an individual component taking account of friction, specific enthalpy, and heat exchange (gas/liquid, wall/gas, liquid/gas, wall/liquid) in order to simulate the gas–liquid flows in a well and a gas-gathering network.

The Peng–Robinson equation of state (modified Van der Waals equation) was used as the equation of state, where $p$ is pressure, $T$ is temperature, $V$ is molar volume, $R$ is the gas constant, and $a$ and $b$ parameters are the coefficients responsible for the forces of attraction between molecules and the finite volume of the molecules calculated through the parameters at the critical point:

$$p = \frac{RT}{V - b} - \frac{a\alpha}{V(V + b) + b(V - b)} \tag{1}$$

.

It is necessary to take into account the temperature dependence of the a and $b$ coefficients. If compressibility factor Z is introduced, then the Peng–Robinson equation of state can be represented as a cubic equation of Z. After solving this equation, the maximum root was selected for the gas phase, while the minimum root was selected for the liquid phase. Additional ratios are required to determine a and b coefficients and the molar mass for a mixture of several components, which are called blend recipes (Soave ratios).

The model for calculating the phase state of the mixture allows obtaining the quantities (in moles) of the hydrocarbon liquid and gas coexisting in a tank or vessel at a given pressure and temperature. These calculations also make it possible to define the composition of the existing liquid and gas hydrocarbon phases.

In a multicomponent system, the separation of the components between the liquid and gas phases is described by the equilibrium ratio (coefficient) for the given component.

The equilibrium coefficient is defined as the ratio of the molar fraction of the component in the gas phase to the molar fraction of the component in the liquid phase. At low pressures, Dalton's law is applicable to a mixture of gases, and Raoult's law is applicable to

regular solutions. Raoult's law allows the expression of the partial pressure of a component in a solution through the pressure of saturated vapors of the given component. Dalton's law expresses the partial pressure of a component in a gas through the pressure in the system.

This model assumes that the value of the equilibrium coefficients for any component does not depend on the total composition of the mixture, and since the saturation pressure depends only on temperature, the equilibrium coefficients depend only on pressure and temperature in the system.

The temperature of the mixture can be determined if the total enthalpy is known, and the empirical Lohrenz–Bray–Clark formula was used to calculate the viscosities.

The Lee–Gonzalez–Eakin method was used to calculate the gas phase viscosity. This method is a semi-empirical correlation in which the gas viscosity is expressed through temperature, gas phase density, and molar mass. The proposed correlation predicts the viscosity value with a standard deviation of 2.7% and a maximum deviation of 8.99%, which is why this method cannot be used for sulfurous gases.

Calculation of friction with the channel walls for a Newtonian fluid is made using the standard approaches as closing relations of the system.

For a single-phase liquid or gas flow, the friction pressure loss due to the wall friction included in the momentum conservation equations is usually expressed through the specific mass flow rate of the mixture with a density of the medium, while the friction coefficient is a function of the Reynolds number at the fixed channel geometry. Various correlations of the friction coefficient (in laminar or turbulent mode) are used to calculate the pressure loss. Transition flow uses linear interpolation between the expressions for the laminar and turbulent modes.

The situation is more complicated in the case of a two-phase flow. Here, the pressure loss depends not only on the flow rates and Reynolds numbers of the individual components but also on the structure of the two-phase flow. The Lockhart–Martinelli approach is common for calculating friction in two-phase modes: according to it, a pressure loss in a two-phase flow can be calculated as the product of the pressure loss of a single-phase flow of any phase with the flow rate of the two-phase mixture and the two-phase friction multiplier. Simpler, Hagedorn–Brown [2], Duns–Ros [3], and Orkiszewski [4] models can be used to calculate the friction in a two-phase flow. There are also transient two-phase flow models [5,6], but they are not covered in this article.

Heat fluxes coming from the wall into the liquid and gas phases were calculated in order to determine the heat exchange with the wall in the proposed model (heat transfer coefficients for each phase are calculated through the dimensionless Nusselt number).

The proposed mathematical models contribute to the development of the theoretical aspects of time simulation of multiphase media, as well as the research into important practical problems in the industry. In order to make a computational algorithm, all the previous conservation equations were rewritten in a finite-difference form.

*2.2. Choke Model*

In the basic model, a choke is a local abrupt taper of a pipeline and then its abrupt enhancer (Figure 2).

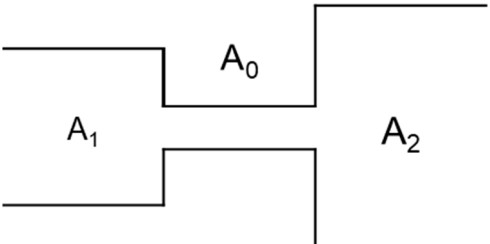

**Figure 2.** Choke model geometry. $A_0$, $A_1$, and $A_2$ are flow areas.

In this case, the pressure loss at the choke is the sum of irreversible losses in case of the abrupt taper of the pipeline from section $A_1$ to section $A_0$, irreversible losses in case of the abrupt enhancer of the channel from section $A_0$ to section $A_2$, and local friction losses in the narrow section of the channel:

$$\Delta p = \xi \frac{\rho u_0^2}{2} = \left( \xi_{taper} + \xi_{enhancer} + \xi_{friction} \frac{l_0}{D_0} \right) \frac{\rho u_0^2}{2} \tag{2}$$

where $\rho$ is the fluid density, $u_0$ is the fluid velocity, $\xi$ is the loss factor, $l_0$ is the choke length, and $D_0$ is its inner diameter.

The friction loss coefficient is calculated using the friction model [7]. The taper and enhancer loss coefficients are calculated according to the ratios in [8].

The taper loss coefficient is calculated as follows:

- at $Re < 10$: $\xi_{taper} = \frac{30}{Re}$
- at $10 \leq Re < 10^4$: $\xi_{taper} = A \cdot B \left( 1 - \frac{A_0}{A_1} \right)$

where $A = \sum\limits_{i=0}^{7} a_i (\lg Re)^i$, $a_0 = -25.12458$, $a_1 = 118.5076$, $a_2 = -170.4147$, $a_3 = 118.1949$, $a_4 = -44.42141$, $a_5 = 9.09524$, $a_6 = -0.9244027$, $a_7 = 0.03408265$.

$B = \sum\limits_{i=0}^{2} \left\{ \left[ \sum\limits_{j=0}^{2} a_{ij} \left( \frac{A_0}{A_1} \right)^j \right] (\lg Re)^i \right\}$, $a_{ij}$ coefficients are presented in Table 1.

**Table 1.** Values $a_{ij}$.

| i/j | $10 \leq \mathbf{Re} \leq 2 \times 10^3$ | | | $2E3 \leq \mathbf{Re} \leq 4 \times 10^3$ | | |
|---|---|---|---|---|---|---|
| | **0** | **1** | **2** | **0** | **1** | **2** |
| 0 | 1.07 | 1.22 | 2.9333 | 0.5443 | −17.298 | −40.715 |
| 1 | 0.05 | −0.51668 | 0.8333 | −0.06518 | 8.7616 | 22.782 |
| 2 | 0 | 0 | 0 | 0.05239 | −1.1093 | −3.1509 |

- at $Re > 10^4$: $\xi_{taper} = \frac{1}{2} \left( 1 - \frac{A_0}{A_1} \right)^{3/4}$

The enhancer loss coefficient is as follows:

- at $Re < 10$: $\xi_{taper} = \frac{30}{Re}$
- at $10 \leq Re \leq 500$:

$$\begin{aligned} \xi_{taper} = {}& 3.62536 + 10.744a - 4.41041a^2 + \\ & + b\left(-18.13 + 56.77855a + 33.40344a^2\right) + \\ & + b^2\left(30.8558 + 99.9542a - 62.78a^2\right) + \\ & + b^3\left(-13.217 - 53.9555a + 33.8053a^2\right) \end{aligned}$$

where $a = \left( 1 - \frac{A_0}{A_2} \right)^2$, $b = \frac{1}{\lg Re}$;

- at $500 < Re < 3.3 \cdot 10^3$:

$$\begin{aligned} \xi_{taper} = {}& -8.44556 - 26.163a - 5.38086a^2 + c\left(6.007 + 18.5372a + 3.9978a^2\right) + \\ & + c^2\left(-1.02318 - 3.0916a - 0.680943a^2\right) \end{aligned}$$

where $a = \left( 1 - \frac{A_0}{A_2} \right)^2$, $c = \lg Re$;

- at $Re \geq 3.3 \cdot 10^3$: $\xi_{taper} = \left( 1 - \frac{A_0}{A_2} \right)^2$.

*2.3. ESP Model*

The ESP model allows for defining the head created by the pump depending on the flow rate of the mixture. Therefore, the momentum conservation equation is replaced by a pressure differential equation:

$$L_c \frac{\partial p}{\partial l} = g\rho_m \Delta H$$

where $L_c$ is the length of the pump stage, $\Delta H$ is the pump head, and $\langle \rho \rangle$ is density of the mixture.

$$\Delta H = \Delta H_B^* K_H (2 - q - A(1-q)^2),\ K_H = K_Q = 1 - \frac{150}{Re},\ K_\eta = 1.08 - 12 Re^{-0.45}$$

$$\eta = \frac{g\langle\rho\rangle\Delta H Q}{\Delta N},\ q = \frac{Q}{Q^* = \frac{Q}{(K_q Q_B^*)}},\ A = 0.66 + 16 Re^{-0.68},\ Re = \frac{4.3 + 0.816 n_s^{0.274}}{n_s^{0.575}} Q_B^* \frac{\langle\rho\rangle}{\langle\mu\rangle} \left(\frac{\omega}{Q_B^*}\right)^{1/3},$$

$$n_s = 193\omega (Q_B^*)^{0.5} (g\Delta H_B^*)^{-0.75}$$

where $K_Q$, $K_H$, and $K_\eta$ are the supply, head, and efficiency coefficients; $\omega$ is the angular speed of the shaft rotation; $\langle\mu\rangle$ is the effective viscosity of the mixture; and $n_s$ is the speed coefficient.

$$Q_B^* = Q_{BH}^* \frac{\omega}{\omega_H},\ \Delta H_B^* = \Delta H_{BH}^* \left(\frac{\omega}{\omega_H}\right)^2,\ \Delta N_B^* = \Delta N_{BH}^* \left(\frac{\omega}{\omega_H}\right)^3$$

where $Q_{BH}^*$, $\Delta H_{BH}^*$, and $\Delta N_{BH}^*$ are the motor stage characteristics at the rated angular velocity $\omega_H = 50$ Hz.

## 3. Method of Automatic Adjustment to the Actual Data

The number of measured parameters of wells in the existing fields is less than necessary for a comprehensive description of the physical system. In order to adjust the model intended for forecasting and calculation, the unknown parameters should be as close as possible to the actual ones. An automatic adaptation algorithm based on neural networks was used to identify any hidden dependencies between the measured and unmeasured values.

The adaptation algorithm consists of two main modules: a decision-making module and an assessment module. The assessment module is used to approximate the aggregate data specified by the objective function based on the data calculated by the hydrodynamic simulator (Figure 3), whose computational kernel performs a deterministic calculation of the multiphase fluid movement in the reservoir and the well [9]. The objective function can be set arbitrarily based on the measured parameters and the simulator output so that all values are within the $[-1, 1]$ range, where higher values correspond to higher accuracy. The decision-making module was used to determine the values of the adjusted parameters of the simulator based on the approximation performed by the assessment module.

Since direct training of the network requires knowledge of the optimal set of parameters for each set of the input values, the network was trained by end-to-end gradient propagation through the assessment module (Figure 4) by the chain rule formula:

$$\frac{\delta v}{\delta w} = \frac{\delta v}{\delta a} \cdot \frac{\delta a}{\delta w} \tag{3}$$

where $v$ is the output value of the assessment module, $a$ is the vector of the adjusted parameter values, and $w$ is the internal variables of the decision-making model.

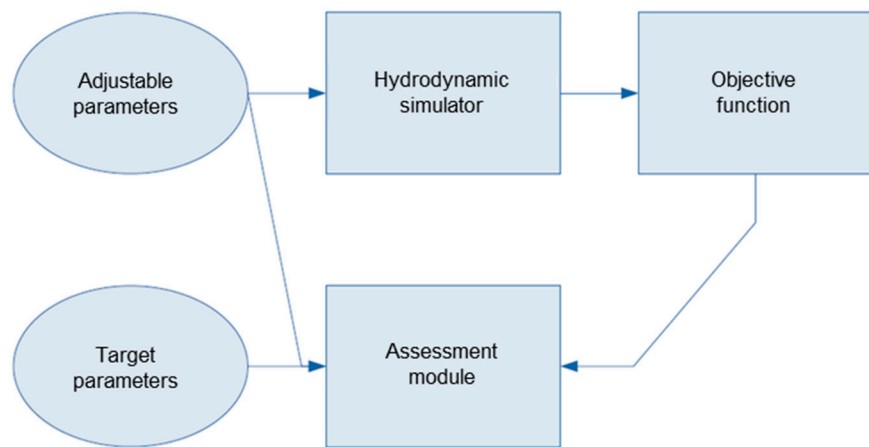

**Figure 3.** Assessment module training.

The architecture of a fully connected two-layer neural network (Figure 5) with RectifiedLinearUnit as an activation function was chosen for the decision-making module. The output layer uses the hyperbolic tangent activation function. The network receives measurement data from a real well as input. The network provides the values of the parameters sent to the simulator as its output.

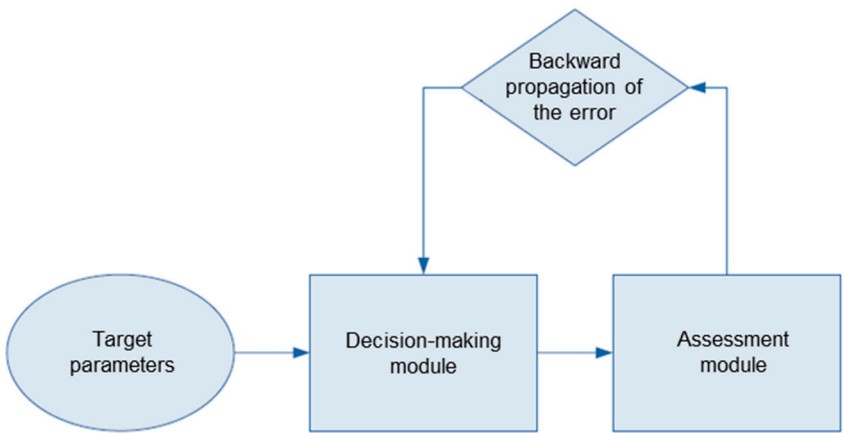

**Figure 4.** Decision-making module training.

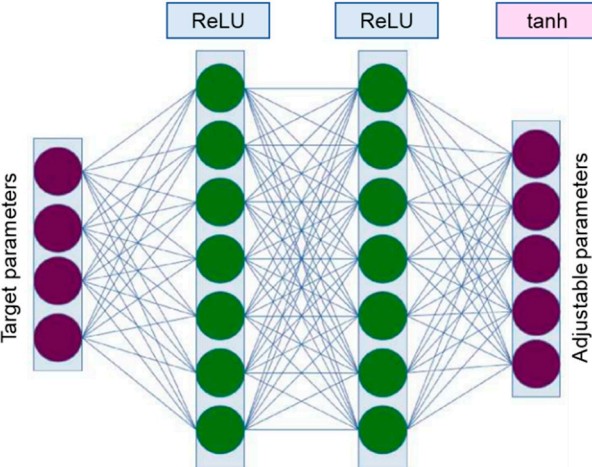

**Figure 5.** Architecture of the neural network for the decision-making module.

This calculation can be performed directly since both modules were implemented on the basis of the TensorFlow [10] computation graph. Adam from the TensorFlow package was used as an optimizer that implements the backpropagation of the error algorithm based on the calculated gradient value.

The architecture of a fully connected three-layer neural network (Figure 6) with RectifiedLinearUnit as an activation function was chosen for the assessment module. The output layer uses a single neuron with the hyperbolic tangent activation function. The network receives measurement data from a real well and the vector of the simulator parameters as input. The network provides the predicted value of the objective function as its output. RectifiedAdam from TensorFlow Addons package with the mean-square deviation as a loss function was used as an optimizer.

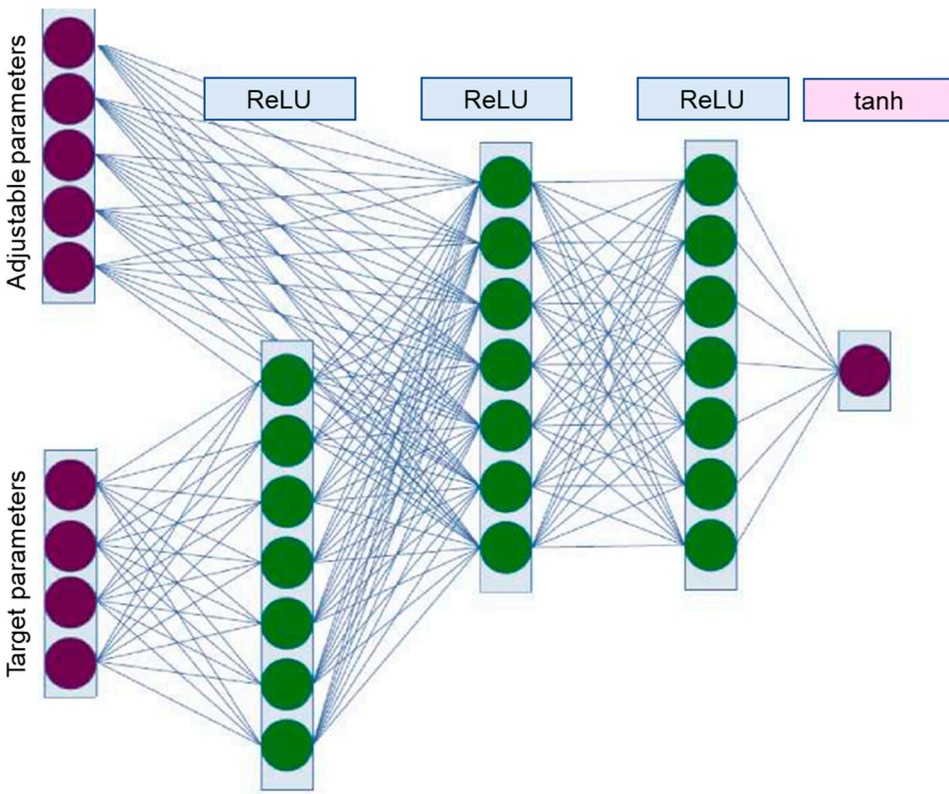

**Figure 6.** Architecture of the neural network for the assessment module.

### 3.1. Problem Statement

Two tasks were considered in this paper: validation of the correctness of the developed simulator and assessing the quality of the automatic adaptation algorithms.

### 3.2. Validation of Correctness

In order to validate correctness, it is necessary to compare with a reference, with the PIPESIM simulator used as one.

A deviated well (Table 2) with a maximum deflection angle of about $32°$ was adopted as the input parameters for the model in PIPESIM software and the methodology used.

**Table 2.** Well geometry.

| Depth along the Wellbore, m | Azimuthal Angle, ° |
|:---:|:---:|
| 0 | 1.9 |
| 270 | 2.0 |
| 400 | 0.9 |
| 820 | 0.7 |
| 950 | 0.9 |
| 2070 | 3.2 |
| 2220 | 11.9 |
| 2340 | 24.4 |
| 2470 | 29.0 |
| 2620 | 32.1 |
| 2770 | 28.5 |
| 2910 | 26.2 |
| 3060 | 28.9 |
| 3200 | 28.4 |
| 3350 | 23.8 |
| 3460 | 11.6 |
| 3580 | 1.3 |
| 3880 | 0.0 |

Fluid properties table (Table 3):

**Table 3.** Fluid properties.

| Properties | Density, kg/m$^3$ |
|:---:|:---:|
| Gas | 0.8087 |
| Liquid | 765.8 |

The following table (Table 4) descibe boundary conditions were used for calculation:

**Table 4.** Reservoir conditions.

| Pressure, Bar | Mixture Temperature, K | OGR | Watercut, % |
|:---:|:---:|:---:|:---:|
| 422 | 378.5 | 0.000715 | 0 |

Beggs and Brill's methods were used to calculate the parameters in the well in the author's simulator and in PIPESIM software.

Two PVT models were used to compare the calculation results: the black oil model and the compositional model. Compositional model was: $N_2$, $CO_2$, C1, C2, C3, C5+, C7+, C11+, C19+, and C36+.

There was no surface or bottom hole equipment such as choke or ESP.

The graphs of the distribution of the parameters of pressure, temperature, gas mass rate, and gas volume gate (Figures 7–10) along the wellbore are shown below.

As mentioned above, the proposed model of the gas–liquid flow is unsteady. Therefore, the steady flow (stationary mode) (Figure 11) is adopted as a comparison of the method developed with PIPESIM.

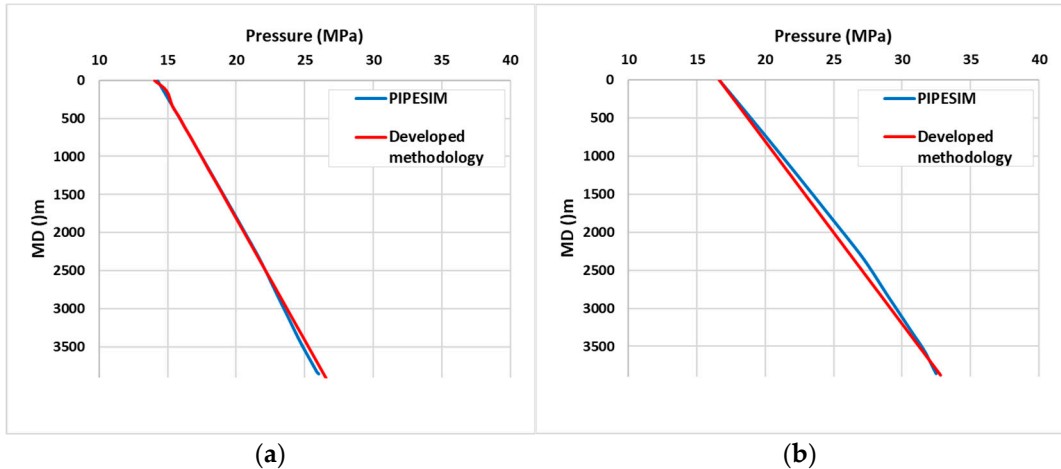

**Figure 7.** Pressure distribution along the wellbore. The blue line represents the result of the PIPESIM calculation, while the red line represents the result of the calculation of the proposed model. The black oil model is (**a**), while the compositional model is (**b**).

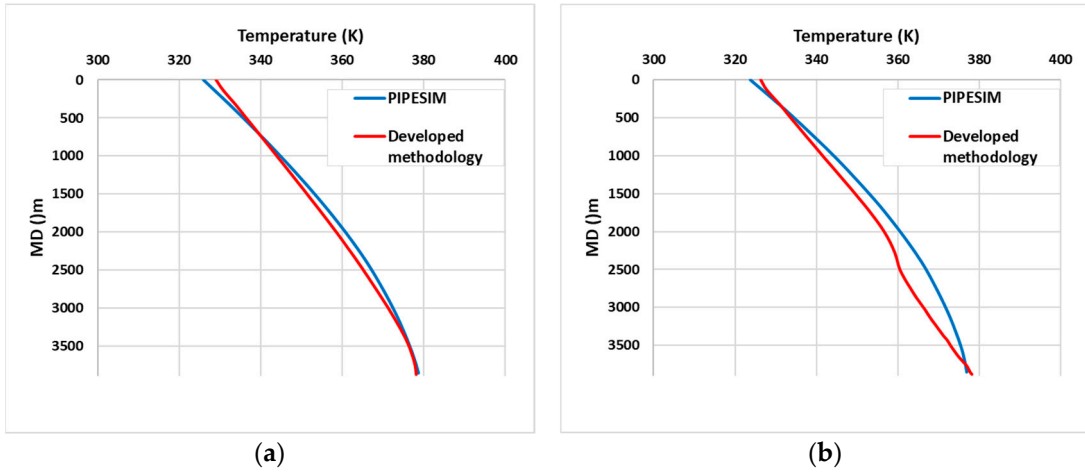

**Figure 8.** Borehole wall temperature distribution along the wellbore. The blue line represents the result of PIPESIM calculation, while the red line represents the result of calculation of the proposed model. The black oil model is (**a**), while the compositional model is (**b**).

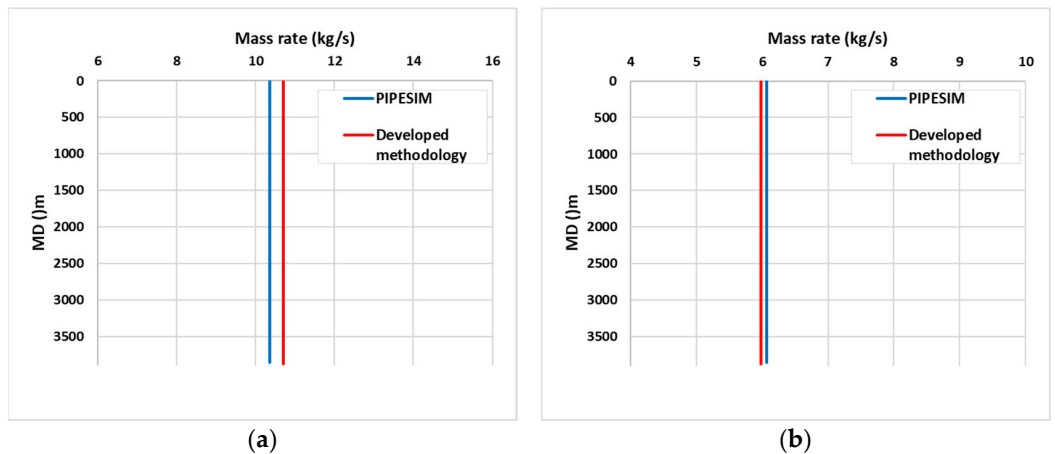

**Figure 9.** Mass gas flow rate distribution along the wellbore. The blue line represents the result of PIPESIM calculation, while the red line represents the result of calculation of the proposed model. The black oil model is (**a**), while the compositional model is (**b**).

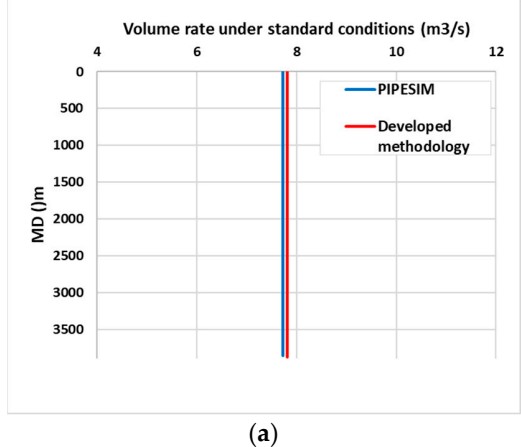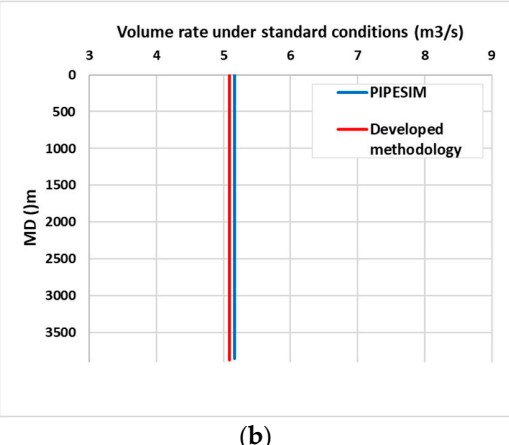

**Figure 10.** Volume gas flow rate distribution (standard condition) along the wellbore. The blue line represents the result of PIPESIM calculation, while the red line represents the result of calculation of the proposed model. The black oil model is (**a**), while the compositional model is (**b**).

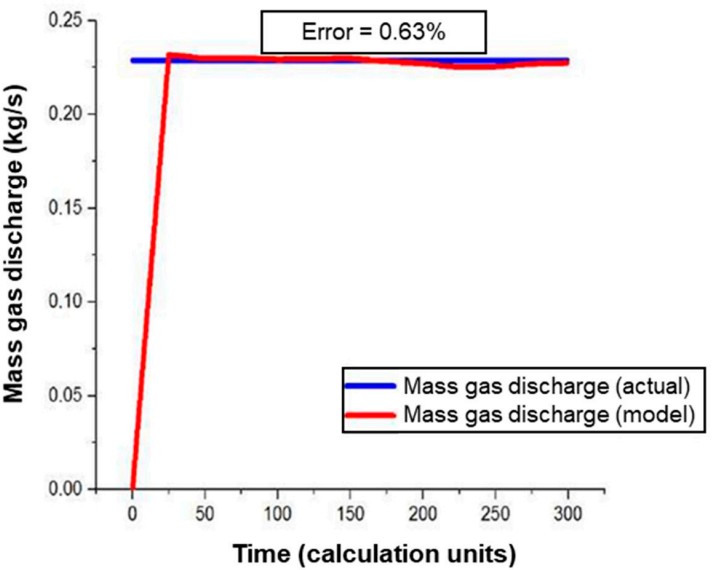

**Figure 11.** Adjustment of the model to actual values in the virtual flow meter mode.

In addition to the above parameters, the proposed method allows calculating the parameters of density, velocity, volume fraction, and viscosity of each phase (liquid, water, gas), as well as the mass, density, and inner radius of the layer of paraffin formed at any time.

### 3.3. Automatic Adjustment Quality

Quality was assessed based on two criteria: velocity and accuracy.

The velocity of the algorithm was determined by the speed of training and the speed of picking. In the tests carried out on a PC with the following configuration: Intel Core i5-7600 CPU at 3.5 GHz and 12 GB memory, the training took 5–7 min of real time, while the time of the calculation itself was about 3 s.

In order to assess the accuracy of the adaptation algorithm, it is necessary to define the accuracy of determining the measured parameters. In case of adjustment to actual values, there is no need to achieve a mistie less than the measurement accuracy. A 5% accuracy for the target parameters was used in this paper.

A set of input and output data divided into the training and test sets was used to assess the quality of the automatic adjustment. In the case of the training set, the algorithm was adjusted to the target parameters with an accuracy of 5%, then the adjusted model

parameters were sent to the test set, and the target values were compared. The quality of the adjustment was considered acceptable if the mistie in the test set did not exceed 5%.

A table (Table 5) comparing the target values on the training and test sets is shown below.

**Table 5.** Comparison of the target parameters on the training and test sets.

| Parameter | Training Set | | | Test Set | | |
|---|---|---|---|---|---|---|
| | Simulator | Experiment | Error, % | Simulator | Experiment | Error, % |
| Volume oil fraction | 6.5 (%) | 6.3 (%) | 3.3 | 5.9 (%) | 6.2 (%) | 4.8 |
| Volume gas fraction | 93.5 (%) | 93.7 (%) | 0.2 | 94.1 (%) | 93.8 (%) | 0.3 |
| Mass oil flow rate | 0.87 (kg/s) | 0.91 (kg/s) | 4.4 | 0.85 (kg/s) | 0.82 (kg/s) | 3.5 |
| Mass gas flow rate | 20.12 (kg/s) | 19.67 (kg/s) | 2.23 | 19.12 (kg/s) | 19.93 (kg/s) | 4.0 |

## 4. Discussion

By using the example of modeling a part of an oil-gathering network, which includes a number of sensors and measurement systems, this paper presented the concept of a virtual measurement system based on a hydrodynamic simulator and a deterministic model of the measurement system. The concept was tested on real experimental data and applied in real time. Acceptable accuracy of predictions of gas and oil discharge rates was obtained. Differences in calculated parameters are related to the difference in viscosity models [9,11].

It was found, based on the analysis, that the relations used to calculate friction of the multiphase flow with the borehole walls are one of the main sources of uncertainty. An exhaustive search of the typical friction models (in addition to the Beggs and Brill model used in the calculations) with the remaining parameters fixed showed a spread of the mass flow rate values most likely related to the use of the models outside the scope of their application. The effect of the pipe wall roughness and the gas compressibility parameter on the flow rate—due to its significant volume fraction of more than 89%—was also shown. Thus, it is necessary to measure the properties of the multiphase mixture and to know the process and design features of the well, as well as to understand the ranges within which the measured parameters might vary in order to use the developed methods in real situations to improve the prediction accuracy.

Further development of the methods will allow for fundamentally new opportunities to improve the measurement systems created on the principle of multiple run metering systems to appear, whose virtualization multiplies the density and reliability of data, to practically implement the principles described in the above-mentioned patents, and to significantly improve the quality of the information base for further development of the smart field technologies.

## 5. Results and Conclusions

The results of the operation of the previous proprietary unsteady influx simulator were compared with the results of the PIPESIM simulator based on actual data. The comparison showed a satisfactory fit for a number of target parameters, such as pressure, mass flow rate, and temperature.

An algorithm for the automatic adaptation of the model to actual data was developed and tested. The selected algorithm architecture allows quick fine-tuning and retraining neural networks to adjust to certain data types: GOR and WOR parameters of a well, typical flow rates, borehole geometry, etc.

Assessment of the quality of adjustment showed that it is possible to achieve the required level of reliability subject to a sufficient amount of data available in the training set. However, further research using actual data is needed to determine the limits of the applicability of this algorithm.

**Author Contributions:** R.V.: investigation and writing—original draft preparation; A.C.: writing—original draft preparation, writing—review and editing, and conceptualization; B.K.: visualization,

software, and writing—review and editing; P.L.: conceptualization and writing—review and editing; R.K.: formal analysis and writing—original draft preparation; M.K.: writing—review and editing; E.U.: methodology and resources; V.U.: resources and supervision. All authors have read and agreed to the published version of the manuscript.

**Funding:** This research received no external funding.

**Informed Consent Statement:** Not applicable.

**Data Availability Statement:** Not applicable.

**Conflicts of Interest:** The authors declare no conflict of interest.

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
