# Peer review of "Development of Virtual Flow-Meter Concept Techniques for Ground Infrastructure Management"

_energies, doi:10.3390/en16010400_

Round 1

Reviewer 1 Report

Review of the article

Development of virtual flow-meter concept techniques for ground infrastructure management

The article is relevant in the context of the development of hydrocarbon deposits, because the efficiency of its extraction depends on many factors. One of these factors is the determination of the parameters of the two-phase flow in a well and onshore oil and gas gathering lines. It is necessary to know these parameters in order to choose a rational well operation mode.

The paper presents a model of unsteady multiphase flow in a channel. The concept of a virtual measuring system based on a hydrodynamic simulator and a deterministic model of the measuring system is presented. Comparison of calculations by the proposed model with the results of calculations on the PIPESIM simulator is carried out.

In general, the work can be assessed positively.

Comments

1. It is necessary to increase the "Introduction" section, indicating in it the works devoted to the flow of multiphase mixtures in wells. For example, consider the following works:

1) Li X., Fatt Y.Y., Goharzadeh A., Chai J.C., Zhang M. Numerical prediction of deposition in two-phase flow in vertical pipes // International Journal of Heat and Technology. 2021. Vol. 39, No. 1. P. 73-88.

2) De Lorenzo M., Lafon P., Di Matteo M., Pelanti M., Seynhaeve J.-M., Bartosiewicz Y. Homogeneous two-phase flow models and accurate steam-water table look-up method for fast transient simulations // International Journal of Multiphase Flow. 2017. Vol. 95. P. 199-219.

3) Clerc S. Numerical Simulation of the Homogeneous Equilibrium Model for Two-Phase Flows // Journal of Computational Physics. 2000. Vol. 161, No. 1. P. 354-375.

4) Choi J., Pereyra, E., Sarica, C., Lee, H., Jang, I.S., Kang, J. Development of a fast transient simulator for gas-liquid two-phase flow in pipes // Journal of Petroleum Science and Engineering .2013. Vol. 102. P. 27-35.

2. The paper presents the Choke model and ESP model. However, it is not clear, when comparing your calculations with the results of the PIPESIM simulator, are these models taken into account?

3. The paper briefly lists the approaches that are used to calculate pressure losses in a wellbore. It is indicated that several different correlations are implemented in the program code. Beggs&Brill correlation was used to demonstrate the effectiveness of the methods when compared with PIPESIM. Why, when using the same correlation, there are differences in the calculation results? An explanation is required.

4. It is required to indicate the composition of the fluid for which the model was matched.

5. There are errors in the text. On page 7, line 231, table 2 should be indicated. The sentence (p. 2, lines 50-52) is not clear: "The first is used to make a forecast of the simulator output parameters based on the output parameters". In my opinion, instead of Fig. 10 (p. 8, line 243) should be Fig. 7-9. Instead of Fig. 9 (p. 8, line 244) should be Fig. 7.

Author Response

Good day, thank you for comments:

1. It is necessary to increase the "Introduction" section, indicating in it the works devoted to the flow of multiphase mixtures in wells. For example, consider the following works:

1) Li X., Fatt Y.Y., Goharzadeh A., Chai J.C., Zhang M. Numerical prediction of deposition in two-phase flow in vertical pipes // International Journal of Heat and Technology. 2021. Vol. 39, No. 1. P. 73-88.

2) De Lorenzo M., Lafon P., Di Matteo M., Pelanti M., Seynhaeve J.-M., Bartosiewicz Y. Homogeneous two-phase flow models and accurate steam-water table look-up method for fast transient simulations // International Journal of Multiphase Flow. 2017. Vol. 95. P. 199-219.

3) Clerc S. Numerical Simulation of the Homogeneous Equilibrium Model for Two-Phase Flows // Journal of Computational Physics. 2000. Vol. 161, No. 1. P. 354-375.

4) Choi J., Pereyra, E., Sarica, C., Lee, H., Jang, I.S., Kang, J. Development of a fast transient simulator for gas-liquid two-phase flow in pipes // Journal of Petroleum Science and Engineering .2013. Vol. 102. P. 27-35.

Done. We add more recent papers to references list. Among suggested examples there are no closely related subjects, since they describe water-vapor flows or flows contain solid debris, so we use different papers.

  1. The paper presents the Choke model and ESP model. However, it is not clear, when comparing your calculations with the results of the PIPESIM simulator, are these models taken into account?

Done, In the given example there are no choke or ESP. This information is added to paper.

3. The paper briefly lists the approaches that are used to calculate pressure losses in a wellbore. It is indicated that several different correlations are implemented in the program code. Beggs&Brill correlation was used to demonstrate the effectiveness of the methods when compared with PIPESIM. Why, when using the same correlation, there are differences in the calculation results? An explanation is required.

Done. In paper pressure losses calculation briefly described in Methodological approaches section. This approach uses custom viscosity models, which described in detail in author’s previous papers [1,2]. This also explain differences in pressure loss. This information is added to paper.

4. It is required to indicate the composition of the fluid for which the model was matched.

The fluid model consisted of 10 components: N2, CO2, C1, C2, C3, C5+, C7+, C11+, C19+, C36+.

5. There are errors in the text. On page 7, line 231, table 2 should be indicated. The sentence (p. 2, lines 50-52) is not clear: "The first is used to make a forecast of the simulator output parameters based on the output parameters". In my opinion, instead of Fig. 10 (p. 8, line 243) should be Fig. 7-9. Instead of Fig. 9 (p. 8, line 244) should be Fig. 7.

Done. Sentence: “The first is used to make a forecast of the simulator output parameters based on the output parameters.” Has typo, correct version is: “The first is used to make a forecast of the simulator output parameters based on the input parameters.” Result presentation order we consider reasonable.

Reviewer 2 Report

The paper is well written, and the authors have good technical knowledge related to the content in the paper. The use of fluid dynamics with regards to multiphase (gas and liquid) in oil and gas is a significant topic and addition of digital twin concept with neural networks/machine learning is a plus. 

Please check minor formatting and reference errors in the draft (Eg. line 59). 

More details and data sets can be added to support the work when comparing the results with existing commercial simulation software for the developed model and virtualization.

Most references are very old. Additional new references may be added to the list for completeness of the topic.

Author Response

Good day, thank you for your comments:

Please check minor formatting and reference errors in the draft (Eg. line 59). 

Done

More details and data sets can be added to support the work when comparing the results with existing commercial simulation software for the developed model and virtualization.

Done. We added comparison with PipeSim

Most references are very old. Additional new references may be added to the list for completeness of the topic.

Done. New references were added.

    1.  

Round 2

Reviewer 1 Report

There are two tables in the article text with the same number 1 (line 145 and line 234). The article text does not describe fig. 7 and 8. The article text first discusses fig. 5, then fig. 4. Also, first fig. 10, then fig. 9. All these comments needs to be corrected or explained.

Author Response

We corrected all your comments.